# Chloride Penetration of Recycled Fine Aggregate Concrete under Drying–Wetting Cycles

**DOI:** 10.3390/ma16031306

**Published:** 2023-02-03

**Authors:** Chunhong Chen, Lei Wang, Ronggui Liu, Jiang Yu, Hui Liu, Jinlong Wu

**Affiliations:** 1School of Urban Construction, Changzhou University, 21 Gehu Middle Road, Wujin District, Changzhou 213164, China; 2Faculty of Civil Engineering and Mechanics, Jiangsu University, Zhenjiang 212013, China; 3State Key Laboratory of Silicate Materials for Architectures, Wuhan University of Technology, Wuhan 430070, China

**Keywords:** aggregate replacement, chloride content, compressive strength, drying–wetting cycling, mineral admixture, porosity

## Abstract

Recycled fine aggregate (RFA) produced from concrete waste is commonly used in the construction industry; however, its use for structural concrete members has not been extensively studied. Moreover, its durability in a drying–wetting cycle environment still needs to be examined. In this study, the intrusion process of chloride in concrete under the drying–wetting cycles is experimentally characterized. Chloride penetration tests are carried out on concrete with the incorporation of different RFA replacement rates and mineral admixtures (i.e., fly ash and silica fume). The results show that the chloride penetration of recycled fine aggregate concrete (RFAC) is dependent upon the performance of the concrete itself, while the deterioration of chloride ion erosion resistance is due to the combined action of the replacement rate of RFA and the drying–wetting cycles. The incorporation of RFA degrades the properties of RFAC owing to its drawbacks in the degradation of interfacial properties of RFAC. Exposure to the drying–wetting cycle environment causes the content of free chloride ions in RFAC to increase initially before decreasing with the erosion depth, thereby showing an obvious convection zone and diffusion zone. The incorporation of the mineral admixture can effectively improve the compactness of the concrete microstructure and make concrete less susceptible to chloride ions ingress. RFAC mixed with 15% fly ash and 10% silica fume has a comparable resistance to chloride penetration as a natural aggregate concrete, which is a feasible method for the application of RFA.

## 1. Introduction

Concrete is in great demand worldwide in the construction industry. The natural aggregate used for traditional concrete is expensive and the mining process is environmentally unfriendly. The large amounts of construction waste produced every year cause great environmental stress. Therefore, it is necessary to find alternatives to the aggregate. The production of recycled aggregates using waste concrete is becoming a viable option to both solve the problems of concrete production and become environmentally friendly [1,2]. Therefore, it is one of the best alternatives to the natural aggregate.

Recycled coarse aggregate is increasingly accepted in codes for structural use, while the usage of recycled fine aggregate (RFA) in structural concrete is restricted [3]. Many scholars have studied the properties of RFA and investigated the extent of the utilization of FRA in concrete. The strength and durability of the concrete with high utilization of FRA are affected due to the high porosity of FRA. They concluded that the mechanical properties of recycled fine aggregate concrete (RFAC), with an RFA replacement rate of less than 30%, differ little from the properties of ordinary concrete. When the replacement rate of RFA is high, or even 100%, the mechanical properties of the recycled aggregate concrete (RAC) are, to some extent, lower than those of ordinary concrete [4,5]. Leite et al. [6] observed that the water absorption of RFA is 90.6% higher than that of the natural fine aggregate (NFA). Kirthika et al. [7] inferred that the increase in the replacement of RFA decreases the strength and durability of RFAC, however, the microstructure densification of the FRAC mixture improves the performance of FRAC with an increase in curing. Fan et al. [8] prepared RFACs with RFA obtained from two different sources and found that the strength of the RFACs decreased following an increase in the replacement of RFA and observed that the rough texture and friction of RFA led to variations in the FRAC properties. Pereira et al. [9] and Berredjem et al. [10] observed that the workability and strength of the RFAC decreased as the RFA replacement increased, although the strength of the concrete increased with the addition of SP, even with 100% replacement of RFA. They concluded that the ITZ performance was weakened due to the high porosity of RFA, which reduced the strength of FRAC. Li et al. [11] analyzed the RFA properties of sizes ranging from 4.75 mm to 75 μm and found that the water absorption of RFA increased with a decrease in particle size, owing to the increased water requirement of the higher ratio of surface FRA particles. Existing studies have shown that when RFA completely replaces NFA, the mechanical properties of RFAC with mineral additives or more cement mineral additives are sufficient for structural use [12].

However, the structure is susceptible to chloride intrusion and its durability is greatly affected. As a result, steel bars corrode and concrete cracks, resulting in structural deterioration and advanced failures. Even worse, studies have revealed the negative impacts of recycled aggregate on chloride resistance in concrete prepared with it, showing a direct relationship between chloride permeability coefficient and RFA substitution [13]. Alternatively, Evangelista et al. [14] found that, compared with natural aggregate concrete, the chloride diffusivity of RFAC with 100% RFA increased by about 60%. Mardani [15] and Pedro et al. [16] also found that the chloride diffusivity of RFAC increased as the replacement rate of RFA increased. The increase in RFAC chloride diffusivity is due to its porosity. The presence of the old mortar attached to the surface of the recycled aggregate leads to a significant increase in the total porosity of the concrete, which provides the paths for chloride ingress.

The drying–wetting cycle is the key factor affecting the durability of the concrete exposed to the deicing salt areas and marine environments [17]. Compared with the full-soaking conditions, there are more significant gradients of the chloride ion concentration field and pore fluid saturation field in the concrete exposed to the drying–wetting cycles, and the chloride ion transport rate in the concrete is faster than that in the saturated state. The coupling effect of the drying–wetting cycles and chloride erosion aggravates the performance damage of concrete, seriously affecting the durability of the concrete structure, and shortening its service life. Liu et al. [18] inferred that the pore structure is remarkably transformed by Cl^−^ ingress. Bai et al. [19] observed that the binding capacity of chloride diminishes with the incorporation of the recycled aggregate and increased the chloride diffusion coefficient. Wu et al. [20] observed that the impact of the drying–wetting cycles was largely on the surface of RAC. Adding fly ash can improve the durability of RAC, with 20% working as the best fly ash content for durability.

Although RFA is commonly used in the construction industry, its use as a structural concrete member has not been extensively studied. Moreover, its durability in a drying–wetting cycle environment still needs to be further studied. The mechanical properties and durability of concrete are both influenced by its composition and chemical additives. The influence of the composition is well-known and will not be repeated. The addition of admixtures can make beneficial modifications to the performance of concrete. Therefore, it is feasible to prepare concrete structures with high mechanical properties and chloride resistance. In this case, RFAC with different RFA mass fractions (0%, 25%, 50%, 75%, and 100%) and six types of concrete with various mineral admixtures and 100% RFA were prepared, and its chloride penetration process was investigated using a drying–wetting cycles test. In other words, the influence of the RFA replacement rate on the chloride penetration of RFAC and the influence of mineral additives on improving the chloride resistance of RFAC were studied. Silica fume and fly ash are the most commonly used mineral additives in concrete since they produce better workability and higher chloride resistance in the concrete. The improvement effect of these two mineral additives on the chloride resistance of RFAC was analyzed, and the application of the concrete prepared with RFA in a drying–wetting cycle environment is discussed. The combined effect of the drying–wetting cycle and chloride erosion on the properties of RFAC has not yet been thoroughly studied and represents the main objective of this study.

## 2. Experimental

### 2.1. Materials

Ordinary Portland cement (OPC) and mineral admixtures were used as cementing materials in the study. Two kinds of mineral admixtures, silica fume (S) and fly ash (F), were selected to replace a portion of the cement to improve the performance of RFAC. The chemical compositions of the cement, silica fume, and fly ash are presented in Table 1. Natural coarse aggregate (NCA) is crushed limestone with a particle size of 5–20 mm. Two types of fine aggregates—namely, natural river sand and RFA are used. RFA is produced by concrete waste from the site of an office building demolition. The core strength of the original concrete is 25 MPa. The main properties of the aggregates are shown in Table 2. The performance of the RFA used in this study is satisfactory according to the provisions of the key indicators in GB/T 25176-2010 [21]. In addition, polycarboxylate superplasticizer with a water reduction rate of 25% is used as an additive in the preparation of the concrete.

### 2.2. Mixture Design

A total of 11 concrete mixtures were prepared for this study. On the one hand, four kinds of RFA replacement rates were selected to investigate the influence of the RFA replacement rate on the performance of RFAC: 0%, 25%, 50%, 75%, and 100%, respectively. Mixtures prepared with OPC and RFA were designated as R*α,* where *α* is the RFA replacement rate (0%, 25%, 50%, 75%, and 100%). For example, R0 is the concrete with 0% RFA, which is the natural aggregate concrete and serves as the control group. Furthermore, R25 is the concrete with 25% RFA. On the other hand, mixtures were prepared by 100% RFA and a different number of mineral admixtures to investigate the improvement effect of mineral admixtures on the performance of RFAC, which are designated as RF*β*, RS*γ*, and RF*β*S*γ*, respectively. R, F, and S stand for RFA, fly ash, and silica fume, respectively. Alternatively, *β* and *γ* stand for the mass replacement rate of fly ash and silica fume, respectively. According to previous studies [22,23], two replacement rates of fly ash (10% and 20%) and two replacement rates of silica fume (5% and 10%) were selected in this study. In order to ensure workability, extra water was added to the concrete mixtures because of the water absorption by RFA and was calculated by multiplying the mass of RFA by its water absorption rate. Concrete mixtures were designed for a compressive strength of 40 MPa and the mixture proportions are presented in Table 3. The grading curves of RFA and river sand are shown in Figure 1.

### 2.3. Testing Details

A total of 165 cubes, each 150 mm in size, were cast for compressive strength testing, of which 99 cubes were used for compressive strength testing before the drying–wetting cycles, while 66 cubes were used during the drying–wetting cycles. Simultaneously, 99 cubes, each 100 mm in size, were cast for the porosity test, of which 33 cubes were used for the porosity test before the drying–wetting cycles, while 66 cubes were used during the drying–wetting cycles. A total of 33 specimens with dimensions of Φ100 mm × 50 mm were prepared for the Cl^−^ permeability coefficient test. Meanwhile, 99 cubes, each 150 mm in size, were prepared for a free chloride content test.

After mixing, the specimens were collected after 24 h and transferred to a standard curing room (20 ± 3 °C, RH > 90%) for 28 days. The compressive strength of the concrete is carried out according to the Chinese standard GB/T 50081-2019 [24]. In addition, the early compressive strength of the concrete was also tested. The chloride diffusivity of the concrete was tested based on the Chinese standard GB/T 50082-2009 [25].

The specimens for the drying–wetting cycles are sealed on five sides with an epoxy polyurethane-based coating, with only the top side exposed so as to ensure the diffusion of chloride ions in only one dimension. The drying–wetting cycles test is programmed to investigate the chloride penetration of the concrete. Specimens were first immersed into sodium chloride solution with a 10% concentration for 21 h (as shown in Figure 2), then, placed for 3 h to be surface-dried. Afterward, they were put into an oven that was programmed to 60 °C for 45 h, before being removed and placed for 3 h to cool to room temperature, after which the single drying–wetting cycle was completed. Free chloride ion content was measured for each type of concrete after 12, 24, and 36 drying–wetting cycles. Concrete powders were drilled with an interval of 3 mm approximately 0–24 mm away from the exposed surface. At each location, the concrete powder was collected from three holes and mixed to ensure accurate testing. The powder was crushed until the particle size was less than 0.63 mm and dried in the oven at 105 ± 5 °C for 2 h. Finally, the chloride content was determined according to the Chinese standard GB/T 50081-2019 [24]. The porosity of the concrete was assessed by ASTM C642 after 0, 12, 24, and 36 drying–wetting cycles [26]. The micro morphologies of the concrete were investigated using a scanning electron microscope (SEM).

## 3. Results and Discussions

### 3.1. Compressive Strength

Compressive strength is considered the most significant property of concrete. Figure 3a shows the compressive strength of all types of concrete, at all ages. There is a marked drop in the compressive strength of RFAC, especially at early ages, while the decline in the compressive strength of concrete with a higher RFA replacement rate is greater. After curing for 28 days, the compressive strength of R0 is 45.2 MPa, and the compressive strength of R25, R50, R75, and R100 is 3.3%, 5.8%, 17.0%, and 22.1% lower than that of R0, respectively. Figure 3a also shows that the strength obtained for concrete with mineral admixture and 100% RFA is higher than that of R100 at all ages. Compared with R100, the 28-day compressive strength of RF10, RF20, RS5, and RS10 increases by 4.5%, 10.5%, 7.4%, and 16.5%, respectively, while the 28-day compressive strength of RF10S5 and RF20S10 increases by 19.6% and 26.4%. As observed, the composite incorporation of fly ash and silica fume can improve the compressive strength of concrete more effectively than a single incorporation. In particular, concrete made with 20% fly ash and 10% silica fume provides comparable compressive strength to that of R0. As shown in Figure 3b, a linear relationship is indicated between the 28-day compressive strength of RFAC and the RFA replacement rate. Moreover, the fitting results show that the compressive strength of concrete without a mineral admixture cannot reach 40 MPa when the RFA replacement rate exceeds 57%.

Figure 4 shows the compressive strength of each concrete exposed to 0 and 36 drying–wetting cycles. It should be noted that the compressive strength in Figure 4 exposed to 0 drying–wetting cycles is the 28-day compressive strength in Figure 3. The compressive strength of RFAC with a higher replacement rate of RFA has a greater reduction after 36 drying–wetting cycles. For example, the compressive strength of R0 decreases by 10.4% and that of R100 by 19.6%. The reduction of the RFAC compressive strength with mineral admixtures is smaller than that of R100, while RF20S10 has the smallest compressive strength loss (12.3%) among them, followed by RF10S5 with 14.2%.

An interfacial transition zone (ITZ) is one of the main methods to load transfer in concrete. Compared with natural aggregate concrete, the ITZ of RFAC is weaker due to the presence of old adhered mortar from RFA. Alternatively, the extra water added due to the high-water absorption of RFA would accumulate in the ITZ during the mixing process and affect the performance of the ITZ. There are some inherent microcracks in the ITZ, which propagate continuously due to the high-stress concentration at the crack tips when the concrete is subjected to load. It follows that the presence of weaker ITZ in RFAC results in lower concrete strength compared with natural aggregate concrete. The concrete with a high RFA replacement rate has more ITZ between the aggregate and the old-adhered mortar; thus, its compressive strength is lower. The effect of the drying–wetting cycles promotes the propagation of inherent microcracks in the ITZ and further reduces the compressive strength of RFAC.

### 3.2. Porosity and Cl^−^ Permeability of RFAC

Figure 5 depicts the porosity and Cl^−^ permeability coefficient of RFAC after 28 days of curing. As the replacement rate of RFA increases, both the porosity and Cl^−^ permeability coefficient of the concrete increase gradually. As shown in Figure 5a, the porosity of R100 prepared with 100% RFA is 14%, which is 68.7% higher than that of R0. The incorporation of fly ash or silica fume can effectively reduce the porosity of concrete, especially the composite incorporation of the two mineral admixtures. The porosity of RF10, RF20, RS5 and RS10 is 6.8%, 14.9%, 6.1%, and 16.9% lower than that of R100, respectively, and 56.6%, 39.8%, 57.8%, and 37.3% higher than that of R0, respectively. Conversely, the porosity of RF10S5 and RF20S10 is 22.3% and 33.8% lower than that of R100, respectively, and 24.1% and 8.4% higher than that of R0, respectively. A similar conclusion can be drawn from Figure 5a about the Cl^−^ permeability coefficient of RFAC. The Cl^−^ permeability coefficient of R100 is 1.36 × 10^−12^ m^2^/s, which is twice that of R0. However, the Cl^−^ permeability coefficients of RF10S5 and RF20S10 are 67.6% and 53.2% that of R100, respectively, and 137.3% and 108.1% that of R0, respectively.

This phenomenon further confirmed the negative effect of RFA on the properties of RFAC. In the mixing progress, the addition of extra water increased the actual water–cement ratio of RFAC, which led to the deterioration of the compactness and the high porosity of RFAC. A large amount of old mortar adhered to RFA led to its own high porosity, which also increased the porosity of RFAC. This result further verifies the positive effect of mineral admixtures on the properties of RFAC. As previously detailed [27,28], the appropriate incorporation of fly ash and silica fume can refine the pore structure of RFAC and enhance the compactness, thus, reducing the porosity and Cl^−^ permeability coefficient, which is an effective method to improve the performance of concrete.

The porosity of RFAC is linearly related to the Cl^−^ permeability coefficient, as shown in Figure 5b. The fitting result reveals that the porosity of RFAC is a direct factor, which affects the Cl^−^ permeability coefficient, while concrete with a higher porosity has a worse chloride resistance. This also confirms the previous understanding that the porosity of concrete is not negligible and has a significant impact on the chloride resistance of concrete [29,30]. In the case of the RFAC, an amount of old mortar with a porous structure attaches to the surface of the RFA, which results in a larger porosity of the RFAC in general. This pore structure characteristic makes RAC less permeable and resistant to Cl^−^ ingress than natural aggregate concrete. In addition, the presence of old mortar also weakens the structural and mechanical properties of the ITZ of RFAC. Therefore, RFAC with a looser pore structure is more susceptible to chloride ingress compared with natural aggregate concrete.

The porosity of RFAC exposed to 0 and 36 drying–wetting cycles is shown in Figure 6a, in which the porosity of all concrete exposed to 0 dry–wet cycles is the porosity in Figure 5a. Concrete with a higher displacement rate of RFA has a higher porosity after 36 drying–wetting cycles. For example, the porosity of R0 and R100 reached 11.4% and 20%, respectively, with respective increases of 38.5% and 42.9%, compared with that before the test. Similarly, the porosity of RFAC with a mineral admixture is lower than that of R100, while RF20S10 has the lowest porosity among them, followed by RF10S5. After 36 cycles, the porosity of RF10S5 and RF20S10 is 15% and 12.8%, respectively, which is 25% and 36% lower than that of R100, respectively, and 32% and 12% higher than that of R0, respectively. One particular finding is that the porosity is linearly related to the compressive strength, for all types of concrete exposed to 36 cycles, as observed in Figure 6b. The effect of the drying–wetting cycles causes variation in the pore structures of the concrete. Hydration products, generated by the chemical reaction between the chloride ions and concrete, constantly crystallize, dissolve, and migrate into the pores under the action of the drying–wetting cycles. The original pores of the concrete are gradually divided and filled, which forms new pores, and promotes variation in the pore structure. In addition, the drying–wetting cycles lead to the formation of microcracks in the pore structure of the concrete due to the salt crystallization pressure and osmotic pressure, ultimately, increasing the number of connected pores and eventually showing the coarsening of the pore size. Consequently, the compressive strength of the concrete decreases.

### 3.3. Chloride Penetration of Concrete Exposed to Drying–Wetting Cycles

Figure 7 shows the free chloride contents in the concrete exposed to different drying–wetting cycles. As observed, both the drying–wetting cycles and the replacement rate of RFA show a significant effect on the free chloride contents in concrete. Free chloride content at each location increases as the drying–wetting cycles and RFA replacement rate increase, regardless of the concrete type. After 12 and 36 drying–wetting cycles, free chloride contents for R50 and R100 are compared, where free chloride content of R50 are 0.54% and 0.82% at a depth of 3.0 mm, respectively, and those of R100 are about 0.66% and 1.12%, respectively. Moreover, the free chlorine content of the concrete with a high RFA replacement rate increases faster. For example, the difference between the chloride contents of R0 and R100 at 3 mm is 0.26% after 12 drying–wetting cycles and 0.51% after 36 cycles. Additionally, the concrete incorporated with a mineral admixture has a much lower free chloride content than R100 at each location, especially the concrete incorporated with both fly ash and silica fume. After 36 drying–wetting cycles, the maximum free chloride content of RF20S10 is 0.64%, which is 57.1% of the maximum chloride content for R100; however, it is comparable to the maximum chloride content of R0.

The durability of RFAC depends on the performance of the concrete itself. The incorporation of RFA degrades the properties of concrete because it increases the porosity and degrades the interfacial properties of concrete. The drying–wetting cycle magnifies these drawbacks of RFAC, resulting in higher porosity and worse interfacial properties. In this regard, there are more paths in the recycled concrete for chloride ingress, and the content of chloride ions inside the concrete increases.

These findings indicate that the incorporation of a mineral admixture (i.e., fly ash and silica fume) reduces the accumulation of free chloride at the exposed surface and delays the diffusion of chloride ions into the interior. This phenomenon is likely due to the modified pore structure of the concrete with a mineral admixture, which can lead to the reduction of the effective diffusion coefficient of chloride ions. Consequently, the chloride penetration process in RFAC with a mineral admixture is much slower compared with that of other concretes during the drying–wetting cycles; therefore, the chloride ion concentration in the concrete is lower.

As illustrated in Figure 7, the convection zone for the free chloride is evident in all types of concrete. Free chloride content at the surface layer (approximately > 3 mm) increases sharply and reaches a local maximum of about 3 mm, regardless of 12, 24, or 36 cycles. The presence of a convection zone is the result of the coupling effect of convection and diffusion, which facilitates the chloride penetration from the external surface towards the interior. At depths beyond 3 mm of the exposed surface, the free chloride content in concrete continues to decline. The depth of the RFAC convection zone is about 5 mm under the drying–wetting cycle conditions, although it tends to move towards the interior following an increase in the drying–wetting cycles. For example, the free chloride content of concrete at 6 mm in Figure 7a was significantly decreased, and the depth is obviously beyond the convection zone. However, the free chloride content of RFAC at 6 mm in Figure 7c is basically equivalent to that at the exposed surface, indicating that the depth of the location is also in the convection zone. Appealingly, at 3 mm, the peak curve for the free chloride content in Figure 7c is significantly smoother than that in Figure 7a and b. Probably the real peak is between 3 and 6 mm after 36 drying–wetting cycles, although unfortunately, it is not captured. This is consistent with previous research [31,32]. In the convective region, pore solutions with a high chloride concentration can be formed in the drying period and will accelerate the transport of the chloride ions in the subsequent wetting period, due to capillary suction. In the diffusion zone, the external environment has little influence on the chloride ingress into the concrete; thus, the diffusion process dominates the chloride penetration, resulting in the reduction of free chloride content in the concrete.

The percentage difference *k* of the free chloride content is shown in Table 4. *k* is defined as:(1)$$k=\frac{c_{R0}-c_{f1}}{c_{R0}}$$
where *c*_R0_ is the free chloride content of R0 after 36 drying–wetting cycles and $c_{f1}$ is the free chloride content of other types of concrete after 36 drying–wetting cycles.

Considering that the free chloride content of concrete is small at depths of 18, 21, and 24 mm, it was not included in the table. The free chloride content of R100 at the same depths is the largest in all concrete, while that of R0 is the smallest after the same drying–wetting cycles. As illustrated in Table 4, the free chloride content of the other concretes is higher than that of R0, so the values of *k* are all negative. The value of *k* in the diffusion zone between R0 and the other concretes with different RFA replacement rates is larger than that in the convection zone, indicating that the negative effect of the RFA replacement rate on the diffusion region is greater than that on the convection region. Interestingly, except for RF10S5 and RF20S10, the absolute values of *k* at 9–15 mm for mixtures are almost greater than those at 0–6 mm. This reveals that fly ash and silica fume allow more beneficial modifications to the chloride resistance of the concrete in the convection zone than in the diffusion zone. However, the composite incorporation of the two admixtures has the same improvement effect on chloride penetration resistance in these two respective zones of concrete. In particular, the absolute value of *k* for RF20S10 at all depths is the smallest, and it is inferred that the chloride penetration resistance is close to R0.

### 3.4. SEM Analysis of RFAC

The microstructures of the concrete specimens are the fundamental reason for the performance of concrete. SEM images of RFAC with different RFA replacement rates after 36 drying–wetting cycles are shown in Figure 8.

As shown in Figure 8a, the internal structure of R0 is relatively compact, with only a few micropores and cracks. The fine aggregate and mortar in R0 possess good bonding properties, and the ITZ is very dense with almost no cracks. The bonding between the fine aggregate and mortar in R50 is slightly looser, with cracks between the fine aggregate, while the mortar can be clearly seen and the quality of the ITZ has obviously deteriorated (Figure 9b). In addition, a number of microcracks and pores can be observed. When the specimens are prepared with 100% RFA, there is no evident boundary in the ITZ between the fine aggregate and mortar. Moreover, the pores and microcracks of R100 have gradually expanded, showing that loose and porous concrete has formed (Figure 9c). This phenomenon is attributed to the higher water absorption rate of RFA. The actual water–cement ratio of RFAC is higher than that of the control group (namely R0) due to extra water, which makes the ITZ of RFAC less compact. In conclusion, compared with R0, the incorporation of RFA introduces weak ITZs into RFAC. After the drying–wetting cycles, there are more pores and cracks in the microstructure of RFAC, which is the reason for its poor resistance to chloride penetration. This is consistent with the research findings of Ye [33].

Loose structure and poor quality of ITZs are the essential reason for the low compressive strength and poor chloride penetration of RFAC. The SEM images of RFAC with 100% RFA and a mineral admixture after 36 drying–wetting cycles are shown in Figure 9. As shown in Figure 9a,c, there are a lot of pores and microcracks in the specimens prepared with 10% fly ash or 5% silica fume, respectively. As the content of fly ash or silica fume increases, the microstructure of RFAC shows a dense trend, and there are fewer microcracks and pores in the structures of RF20 and RS10. In addition, the quality of the ITZs gradually improves, and stronger bonding of the fine aggregate and mortar is observed, as shown in Figure 9b,d. The microstructure of RFAC becomes denser when fly ash and silica fume are incorporated simultaneously (Figure 9e,f). There are only a few pores in the adhesive mortar and ITZs, which are filled with hydration products.

The above phenomena are related to the hydration reaction of the concrete, which is promoted by incorporating either fly ash or silica fume. The performance improvement of ordinary concrete by adding the mineral admixtures is mainly due to the pozzolanic reaction, while the mineral admixture has dual effects on the recycled concrete, such as microaggregate filling and the pozzolanic reaction [34]. The microaggregate effect can fill the adhesive mortar and the old interface between the adhesive mortar and aggregate, thus, refining the pores and making the specimen structure more compact. The pozzolanic reaction consumes Ca(OH)_2_ in the interfaces between the aggregate and mortar and generates more C-S-H gels, which enhances the bonding property of the interfaces and improves the interfaces.

## 4. Conclusions

This paper investigates the chloride penetration of RFAC exposed to drying–wetting cycles. The chloride ion content, compressive strength, and porosity of RFAC, with different RFA replacement rates, are explored.

(1)Under the drying–wetting cycling conditions, there are obvious convection zones and diffusion zones for the free chloride profiles. The depth of the convection zone is about 5 mm, and it increases as the number of cycles increases. The negative effect of the RFA replacement rate is greater on the concrete diffusion zone than on the convection zone.(2)The reduction in the compressive strength of RFAC is larger as the number of cycles or the RFA replacement rate increases. The 28-day compressive strengths of R25, R50, R75, and R100 are 3.3%, 5.8%, 17.0%, and 22.1% lower than that of R0, respectively. The compressive strength of RFAC made with 100% RFA decreases by 19.6% after exposure to 36 drying–wetting cycles. The 28-day compressive strength of concrete has a linear relationship with the RFA replacement rate.(3)The incorporation of RFA increases the porosity of RFAC and causes its durability to deteriorate. Concrete with a higher RFA displacement rate has a higher porosity during the drying–wetting cycles. Before the drying–wetting cycles, the porosity of R100 prepared with 100% RFA was 14%, which is 68.7% higher than that of R0, while the porosity of R100 reached 20% after 36 drying–wetting cycles. The porosity of RFAC is linearly correlated with the Cl^−^ permeability coefficient and compressive strength.(4)The incorporation of fly ash or silica fume greatly improved the performance of RFAC, especially the composite incorporation of 20% fly ash and 10% silica fume. The properties of concrete prepared with 100% RFA, 20% fly ash and 10% silica fume are basically comparable to those of natural aggregate concrete, while the resistance to chloride penetration is also comparable, which is a feasible method for the application of RFA.(5)SEM analysis showed that the bonding strength at the ITZ of RFA in RFAC is weaker than that of natural aggregate concrete. After the drying–wetting cycles, there were more pores and microcracks in the microstructure of RFAC, which was the reason for the deterioration of the chloride penetration resistance. The incorporation of mineral admixtures improved the interface and enhanced the performance of the concrete.

## Figures and Tables

**Figure 1 materials-16-01306-f001:**
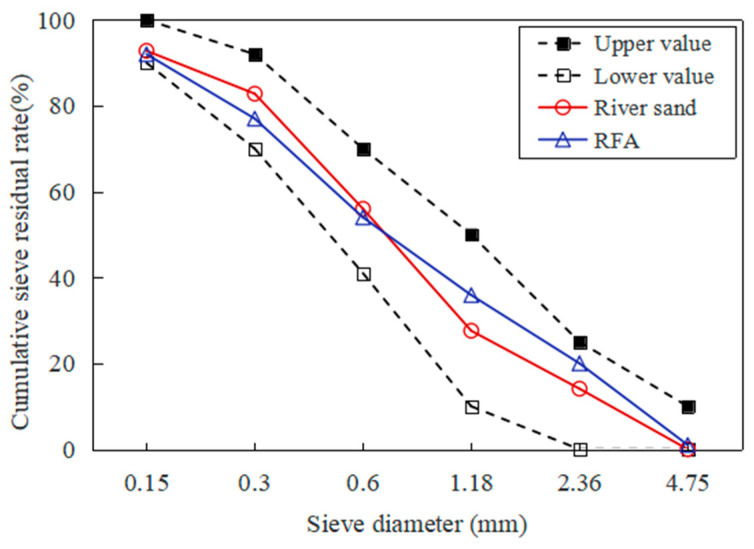
Grading curves of RFA and river sand.

**Figure 2 materials-16-01306-f002:**
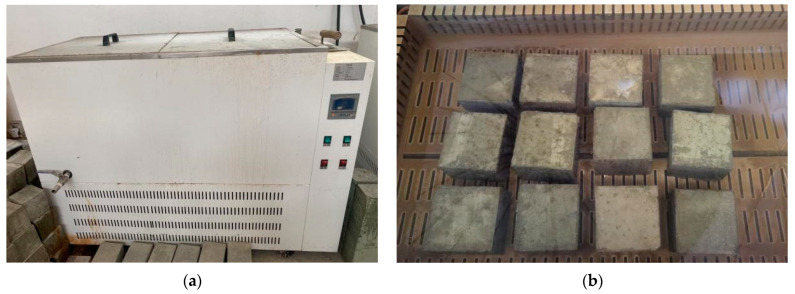
Drying–wetting cycle: (**a**) apparatus for soaking specimens; (**b**) part of specimens immersed in the chloride solution.

**Figure 3 materials-16-01306-f003:**
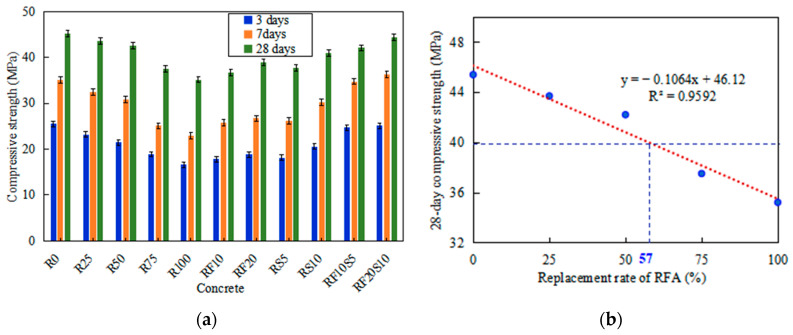
(**a**) Compressive strength of concrete at all ages; (**b**) the fit curve of 28-day compressive strength of concrete without mineral admixture.

**Figure 4 materials-16-01306-f004:**
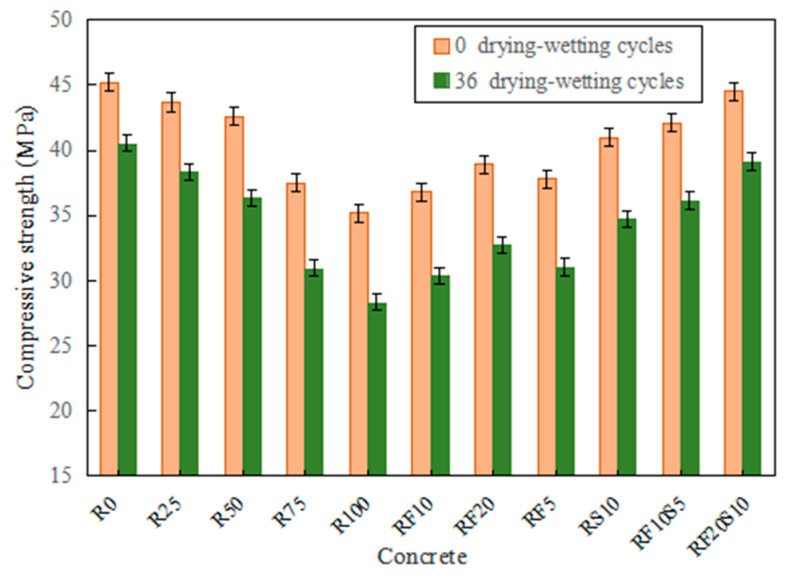
Compressive strength of RFAC exposed to drying–wetting cycles.

**Figure 5 materials-16-01306-f005:**
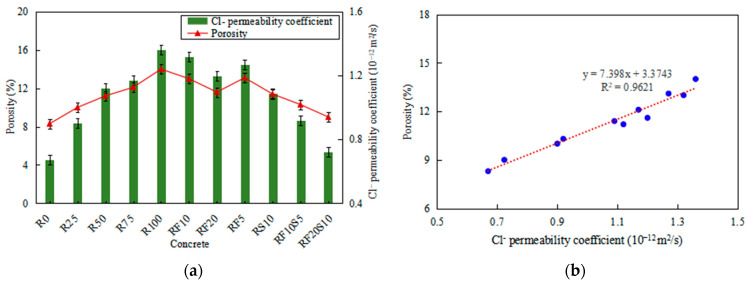
(**a**) Porosity and Cl^−^ permeability coefficients of concrete and (**b**) the fitting curve between them.

**Figure 6 materials-16-01306-f006:**
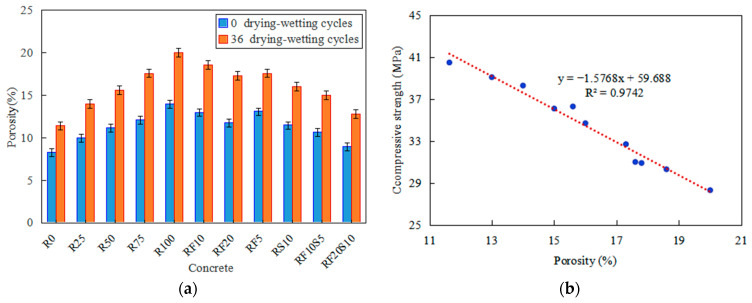
(**a**) Porosity of concrete after drying–wetting cycles; (**b**) a fit curve between compressive strength and porosity of concrete after 36 cycles.

**Figure 7 materials-16-01306-f007:**
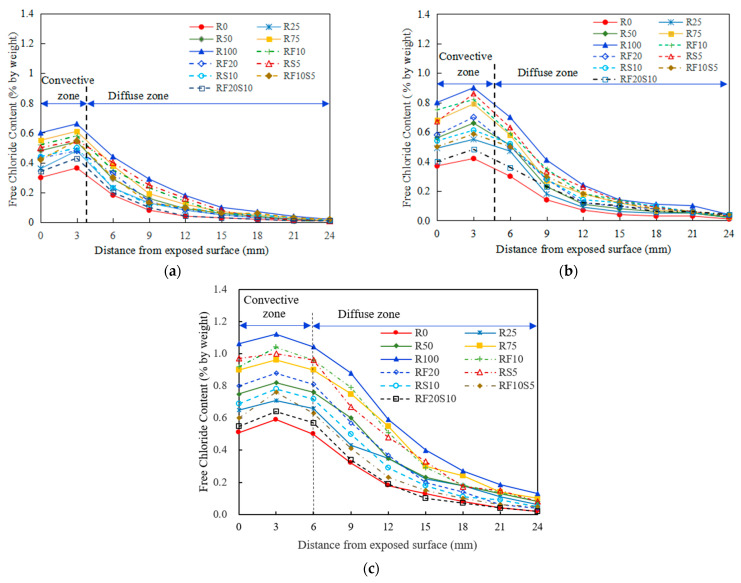
Free chloride distributions in concrete exposed to different drying–wetting cycles: (**a**) 12 cycles; (**b**) 24 cycles; (**c**) 36 cycles.

**Figure 8 materials-16-01306-f008:**
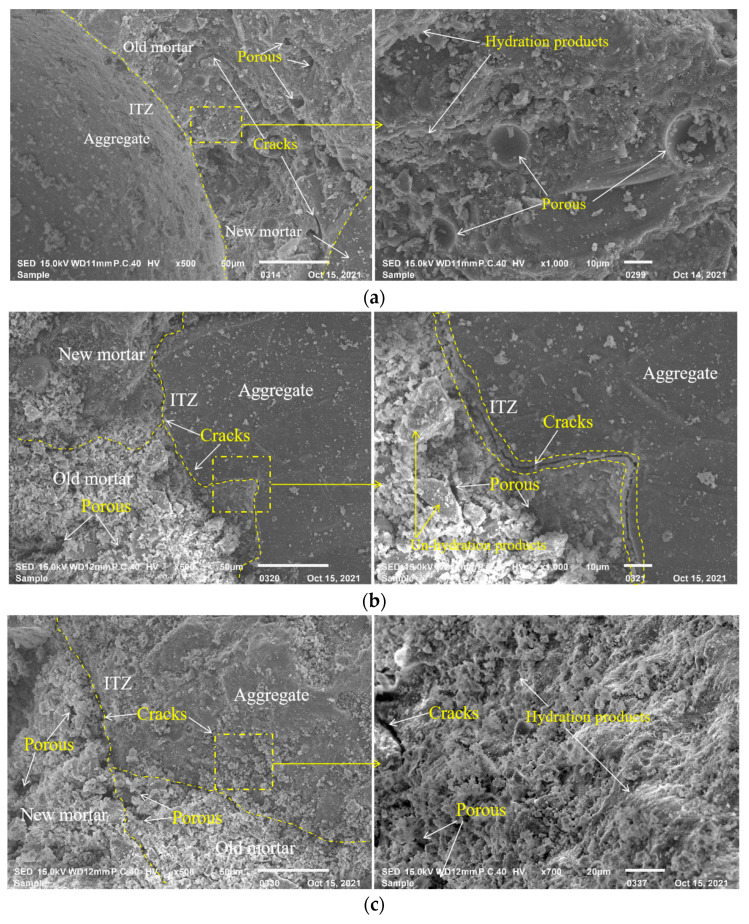
SEM images of specimens after 36 drying–wetting cycles: (**a**) R0; (**b**) R50; (**c**) R100.

**Figure 9 materials-16-01306-f009:**
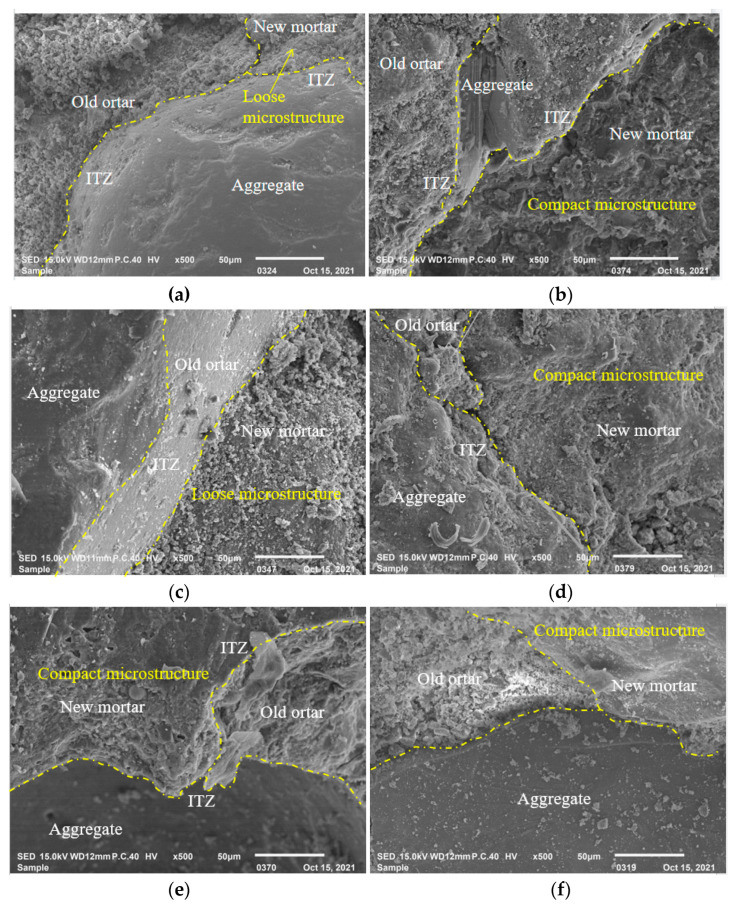
SEM images of concrete with mineral admixture after 36 drying–wetting cycles: (**a**) RF10; (**b**) RF20; (**c**) RS5; (**d**) RS10; (**e**) RF10S5; (**f**) RF20S10.

**Table 1 materials-16-01306-t001:** The chemical compositions of the cementing materials (%).

Materials	SiO_2_	Al_2_O_3_	CaO	MgO	Fe_2_O_3_	SO_3_	Na_2_O	K_2_O	TiO_2_	MnO
OPC	21.22	7.29	60.79	1.8	3.78	1.9	0.28	0.63	0.27	1.17
Silica fume	89.03	4.52	0.37	0.89	0.97	–	–	–	–	0.14
Fly ash	48.3	30.7	4.94	2.53	8.12	0.35	–	0.89	0.79	0.51

Note: “–” is not measured items.

**Table 2 materials-16-01306-t002:** Main properties of the aggregates.

Category	Apparent Density (kg·m^−3^)	Water Absorption (%)	Crush Value	Fineness Modulus
Gravel	2570	2.22	9.7	-
River sand	2586	0.2	11.3	2.4
RFA	2416	11.4	20.4	2.43

**Table 3 materials-16-01306-t003:** Concrete mixture proportions (kg·m^−3^).

Group	OPC	NFA	RFA	NCA	F	S	SP	Water
R0	450	759	0	1034	0	0	2.9	180
R25	450	569	190	1034	0	0	2.9	201
R50	450	375	374	1034	0	0	2.9	222
R75	450	190	569	1034	0	0	2.9	243
R100	450	0	759	1034	0	0	2.9	264
RF10	405	0	759	1034	45	0	2.9	264
RF20	360	0	759	1034	90	0	2.9	264
RS5	428	0	759	1034	0	23	2.9	264
RS10	405	0	759	1034	0	45	2.9	264
RF10S5	383	0	759	1034	45	23	2.9	264
RF20S10	315	0	759	1034	90	45	2.9	264

F: fly ash; S: silica fume; SP: polycarboxylate superplasticizer.

**Table 4 materials-16-01306-t004:** *k* between R0 and other types of concrete.

Depth (mm)	*k* (%)									
	R25	R50	R75	R100	RF10	RF20	RS5	RS10	RF10S5	RF20S10
0	−27.5	−47.1	−76.5	−108.4	−80.4	−56.9	−90.2	−35.3	−17.6	−7.8
3	−20.3	−39.0	−62.7	−89.8	−76.3	−49.2	−66.1	−32.2	−28.8	−8.5
6	−32.0	−52.0	−80.0	−108.6	−92.0	−62.0	−92.0	−44.0	−26.0	−14.0
9	−34.4	−87.5	−134.4	−174.7	−146.9	−78.1	−109.4	−56.3	−28.1	−6.3
12	−94.4	−94.4	−205.6	−227.8	−183.3	−105.6	−166.7	−61.1	−27.8	−5.6
15	−70.5	−78.3	−132.6	−210.1	−126.4	−55.0	−155.8	−39.5	−16.3	−3.9

## Data Availability

The data used to support the findings of this study are available from the corresponding author upon request.

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
