# Peer review of "Chloride Penetration of Recycled Fine Aggregate Concrete under Drying–Wetting Cycles"

_materials, 2023, doi:10.3390/ma16031306_

Round 1
Reviewer 1 Report
The authors are informed to study the difference between target strength as per mix design procedure and Characteristic compressive strength.
Further they are informed to understand the difference between factory made blended cement and site mixed partial replacement.
In addition, the authors are requested to incorporate the reviewer comments in the manuscript.

Author Response
- The reviewer can’t understand the sentence in abstract. “The results show that the deterioration of chloride ion erosion resistance in recycled fine aggregate concrete (RFAC) is the result of various interactions and depends on the performance of concrete itself.” – Please rewrite to understand.
Response: Thank you very much for your comments, which are very helpful for the revision in the abstract. The sentence has been rewritten as follows:
The results show that the durability of recycled fine aggregate concrete (RFAC) depends on the performance of concrete itself, and the deterioration of chloride ion erosion resistance is due to the combined action of the replacement rate of RFA and drying-wetting cycles.
- In Clause 2.1, blended cement is a factory made cement. But in this investigation, the OPC was replaced by Fly-ash and Silica fume. After replacing cement by mineral admixture at mixing point is not called as blended cement. Hence remove the blended cement word and write the sentence suitably.
Response: Our understanding of blended cement is wrong, and this mistake has been fully corrected. In the manuscript, the so-called concrete with blended cement is all replaced by concrete with mineral admixtures.
- In Table 1, the chemical composition of Fly ash seems to be incorrect. Is it Class C type or Class F type? Check the values of FA compositions. SiO2 content may be more than the mentioned value in FA.
Response: Thank you for pointing out the error in the chemical composition of fly ash in the manuscript. It is Class F type in the study. The values of FA compositions have been corrected, as shown in Table 1.
- Fineness modulus of Gravel should be added in Table 2.
Response: As far as I know, the fineness modulus index in Chinese standard is only for fine aggregate, while coarse aggregate has no fineness modulus index, so the fineness modulus of coarse aggregate is not tested in the study. Please correct me if I'm wrong.
- Refer line 97, Target strength 40 MPa is a wrong statement. It is characteristic compressive strength. Target strength should be calculated by considering standard deviation. Modify the sentence suitably.
Response: Thank you for pointing out the error. In the manuscript, the reference to target strength has been deleted and replaced by compressive strength, and the analysis of compressive strength has been modified accordingly.
- Authors are informed to check the mix proportioning of control concrete. The cement content (486 kg/m3 ) is in higher order for M40 grade.
Response: We have checked the mix proportioning of control concrete, and found that the mix proportioning in Table 3 was trial mix proportioning. After trial mixing, the water-cement ratio was adjusted to 0.4, and the mix proportioning in Table 3 have corrected accordingly. We also find that the amount of cement was relatively high.
- Refer line 116, 60 0C can be write as 600C.
Response: Thank you. It has been modified.
- In 115 and 117, 3h can be write as 3 h.
Response: Thank you. It has been modified.
- Fig 2 (b) is a wrong approach, the target mean strength is not 40 MPa.
Response: Thank you for pointing out the error. In the manuscript, the reference to target strength has been deleted and replaced by compressive strength, and the analysis of compressive strength in Fig 2 (b) has been modified accordingly.
- Refer line 142 and 143, “28-day compressive 142 strength of R0 is 45.2 MPa” is not satisfied the target value, since 40MPa is only characteristic compressive strength.
Response: Thank you for pointing out the error in the statement of target strength. In the manuscript, the reference to target strength has been deleted and replaced by compressive strength, and the analysis of compressive strength has been modified accordingly.
- Refer line 145, this is not blended cement. Modify as cement replacement…
Response: Our understanding of blended cement is wrong, and this mistake has been fully corrected. In the manuscript, the so-called concrete with blended cement was all replaced by concrete with mineral admixtures.
- The authors are informed to avoid blended cement and target strength words.
Response: Thank you. These two mistakes have been fully corrected in the manuscript.

Author Response
- Keywords: do not repeat the terms present in the title.
Response: Thank you for your comments. Keywords of the manuscript have been revised accordingly.
- The abstract should clearly indicate the relevance of the work for international research.
Response: Thank you for your comments. We have reviewed the abstract and made some modifications.
- The authors should summarize the central core of knowledge that is the focus of the paper and better discuss its importance.
Response: Thank you for pointing out the shortcomings of the manuscript and the author's insufficient cognition of the knowledge involved in the manuscript. The authors of the manuscript have carefully studied your comments, and have revised the manuscript for a large length according to your comments.
- The literature review has been poorly written. In the literature review part, you should perform a potent literature review and scrutinize the most relevant and recently published papers in high-quality journal articles. The literature review is one of the main parts of a scientific paper to show your novelty, and alert the readers that you are aware of the performed research studies.
Response: Thank you very much for your comments, which are very helpful for the revision in the introduction. The introduction section has been rewritten to describe the latest research results in the field of chloride penetration of recycled fine aggregate concrete under drying-wetting cycles from the following aspects:
- Mechanical properties of recycled fine aggregate concrete and the possibility of application of recycled fine aggregate in concrete
- Effect of recycled fine aggregate on the resistance to chlorine resistance of concrete
- Chloride penetration resistance of recycled fine aggregate concrete in drying-wetting cycling environment
Due to the limited time for revision, many aspects of the research results in the field of chloride penetration of recycled fine aggregate concrete under drying-wetting cycles are not described in the introduction.
- The last part of the introduction should conclude the limitations of the previous studies and provide the main objectives and novelties of this study. You need to clearly address the knowledge gap and provide some meaningful phrases that your study can advance the knowledge and can fill in a knowledge gap that has not been considered yet.
Response: Thank you for pointing out the shortcomings of the introduction. The last part of the introduction has been rewritten according to your comments.
- Research methods should be elaborated and justified.
Response: Thank you. The last part of the introduction has been rewritten according to your comments. The research methods section has been revised to elaborate and demonstrate as much as possible.
- Describe the methods chronologically. This is very important to help the readers to replicate your results. Please cite previous research studies where necessary.
Response: Thank you for your comments. In order to describe the research methods more clearly, chapter 2.3 and chapter 2.4 have been merged and the description of the research methods has been substantially revised.
- The figures have not been appropriately explained as well. The readers cannot perceive the main points.
Response: Thank you for your comments. The interpretation of each figure has been revised. The figures have been analyzed for the purposes of the study, and the main findings have been given.
- The authors should discuss the potential cause of results and not only describe that it happens. In addition, the results should be discussed more deeply in respect to other studies.
Response: Thank you for your comments. A large number of modifications have been made to the research results, and a more in-depth analysis has been carried out. Due to the limited time for revision and limited capacity of the authors, there may be some deficiencies in discussing the results with other studies in more depth.
10.The authors must work harder in the explanation of the results since in work, they found very interesting data that must be discussed in greater depth.
Response: Thank you very much. The authors of the manuscript have carefully studied your comments, conducted an in-depth analysis of the experimental data and phenomena again, and made a large number of revisions to the manuscript according to the analysis results.
11.Grammar and syntax must be improved.
Response: There are indeed many problems in the grammar and syntax. The whole manuscript has been reviewed for many times, and a large number of modifications have been made to the manuscript.

Reviewer 3 Report
Dear Authors,
thank you very much for providing a very interesting article.
The thematic focus of the article is appropriate for the chosen journal.
Chloride penetration with respect to conventional freeze cycle testing in concrete for non-traditional concretes is an area commonly and frequently investigated.
I see the contribution of the present article in the evaluation of a new type of concretes.
The structure of the paper is appropriate and fits the assumptions.
Individual sections:
- title and abstract are well understood,
- the introduction is very short by the standards - you should expand on the world's new studies evaluating waste concrete and their observations.
- for example from: 10.1063/5.0000370
and
10.3390/ma15238287
- the basic concept of introducing material composition is very brief - chapter 2.1 should be significantly expanded to include details of individual components.
- The concept of all concrete mixes is not clear - if the first 4 evaluate the ratio between natural and RFA aggregate, why do the next 7 have only RFA and do not include concretes where the ratio is between natural and RFA plus fly ash or silica fume.
- The description of the concrete names is confusing in places,
- the number of samples for each test is not clear.
- the results do not assess the variance of the values and the standard deviation,
- other assessments as well as the use of visual techniques are not bad and are interesting,
- the results are little discussed with other work on the subject.
Author Response
- title and abstract are well understood
Response: Thanks for your encouragement. We have checked the manuscript again and made some modifications to the abstract in order to express it more accurately.
- - the introduction is very short by the standards - you should expand on the world's new studies evaluating waste concrete and their observations. - for example from: 10.1063/5.0000370 and 10.3390/ma15238287
Response: Thank you very much for your comments, which are very helpful for the revision in the introduction. I studied the abstracts of the two papers you recommended and made major revisions to the introduction. The introduction section has been rewritten to include the world's new studies evaluating waste concrete and their observations.
- - the basic concept of introducing material composition is very brief - chapter 2.1 should be significantly expanded to include details of individual components.
Response: Thank you very much for your suggestions. I described the properties and sources of the materials as much detail as possible, including the types of natural coarse aggregate, the sources of recycled fine aggregate and so on.
4.- The concept of all concrete mixes is not clear - if the first 4 evaluate the ratio between natural and RFA aggregate, why do the next 7 have only RFA and do not include concretes where the ratio is between natural and RFA plus fly ash or silica fume.
Response: Thank you. This is indeed a problem, and the concept of all concrete mixes is not clearly explained. The purpose of preparing different types of concrete has been explained in the manuscript.
A total of eleven concrete mixes are prepared for this study. On the one hand, the influence of the replacement rate of RFA on the performance of RFAC is studied. Four kinds of RFA replacement rates are selected in the study, which are 0%, 25%, 50%, 75% and 100% respectively. Mixes prepared with OPC and RFA are designated as Rα, where α is RFA replacement rate (0%, 25%, 50%, 75% and 100%). For example, R0 is the concrete with 0% RFA, which is the natural aggregate concrete and serves as the control group. And R25 is the concrete with 25% RFA. On the other hand, the improvement effect of mineral admixtures on the performance of RFAC is studied. Concrete mixes are prepared by RFA with a replacement rate of 100% and mineral admixtures of different amount, which are designated as RFβ, RSγ and RFβSγ, respectively. R, F and S stand for RFA, fly ash and silica fume respectively. In addition, β、γ stand for the mass replacement rate of fly ash and silica fume, respectively. According to previous studies [22, 23], the replacement rate of fly ash is 10% and 20%, and the replacement rate of silica fume is 5% and 10%. Considering the different water absorption of RFA and river sand, extra water is added to ensure the workability of mixes, which is obtained by multiplying the mass of RFA by its water absorption rate. The mixes are designed for compressive strength of 40 MPa and concrete mix proportions are presented in Table 3. The grading curves of RFA and river sand are shown in Figure 1.
5.- The description of the concrete names is confusing in places,
Response: Thank you. The whole manuscript has been reviewed several times and numerous revisions have been made to the manuscript, including the description of the description of the concrete names.
6.- the number of samples for each test is not clear.
Response: Thank you very much for your comments. The number of specimens for each test has been given in the manuscript, see chapter 2.3.
7.- the results do not assess the variance of the values and the standard deviation,
Response: Thank you very much for your suggestions. Error bars are given on the main figures.
8- other assessments as well as the use of visual techniques are not bad and are interesting,
Response: Thanks for your encouragement.
9- the results are little discussed with other work on the subject.
Response: Thank you.The results of the manuscript have been revised accordingly according to your comments. Due to the limited time for revision and limited capacity of the authors, there may be some deficiencies in discussing the results with other work on the subject.

Round 2
Reviewer 1 Report
The suggestions are incorporated suitably and hence the paper may be considered for publication.

Reviewer 2 Report
The Authors have carried out all the comments given by the reviewer. The manuscript can be accepted for publication as it is.
Reviewer 3 Report
I have no further comments